# Active Nearest-Neighbor Learning in Metric Spaces

**Aryeh Kontorovich**
Department of Computer Science
Ben-Gurion University of the Negev
Beer Sheva 8499000, Israel

**Sivan Sabato**
Department of Computer Science
Ben-Gurion University of the Negev
Beer Sheva 8499000, Israel

**Ruth Urner**
Max Planck Institute for Intelligent Systems
Department for Empirical Inference
Tübingen 72076, Germany

## Abstract

We propose a pool-based non-parametric active learning algorithm for general metric spaces, called MArgin Regularized Metric Active Nearest Neighbor (MARMANN), which outputs a nearest-neighbor classifier. We give prediction error guarantees that depend on the noisy-margin properties of the input sample, and are competitive with those obtained by previously proposed passive learners. We prove that the label complexity of MARMANN is significantly lower than that of any passive learner with similar error guarantees. Our algorithm is based on a generalized sample compression scheme and a new label-efficient active model-selection procedure.

## 1 Introduction

In this paper we propose a non-parametric pool-based active learning algorithm for general metric spaces, which outputs a nearest-neighbor classifier. The algorithm is named MArgin Regularized Metric Active Nearest Neighbor (MARMANN). In pool-based active learning [McCallum and Nigam, 1998] a collection of random examples is provided, and the algorithm can interactively query an oracle to label some of the examples. The goal is good prediction accuracy, while keeping the label complexity (the number of queried labels) low. MARMANN receives a pool of unlabeled examples in a general metric space, and outputs a variant of the nearest-neighbor classifier. The algorithm obtains a prediction error guarantee that depends on a noisy-margin property of the input sample, and has a provably smaller label complexity than any passive learner with a similar guarantee.

The theory of active learning has received considerable attention in the past decade [e.g., Dasgupta, 2004, Balcan et al., 2007, 2009, Hanneke, 2011, Hanneke and Yang, 2015]. Active learning has been mostly studied in a parametric setting (that is, learning with respect to a fixed hypothesis class with a bounded capacity). Various strategies have been analyzed for parametric classification [e.g., Dasgupta, 2004, Balcan et al., 2007, Gonen et al., 2013, Balcan et al., 2009, Hanneke, 2011, Awasthi et al., 2013].An active model selection procedure has also been developed for the parametric setting Balcan et al. [2010]. However, the number of labels used there depends quadratically on the number of possible model classes, which is prohibitive in our non-parametric setting.

The potential benefits of active learning for non-parametric classification in metric spaces are less well understood. The paradigm of cluster-based active learning [Dasgupta and Hsu, 2008] has been shown to provide label savings under some distributional clusterability assumptions [Urner et al., 2013, Kpotufe et al., 2015]. Certain active learning methods for nearest neighbor classification are known to be Bayes consistent [Dasgupta, 2012], and an active querying rule, based solely on information in

the unlabeled data, has been shown to be beneficial for nearest neighbors under covariate shift [Berlind and Urner, 2015]. Castro and Nowak [2007] analyze minimax rates for a class of distributions in Euclidean space, characterized by decision boundary regularity and noise conditions. However, no active non-parametric strategy for general metric spaces, with label complexity guarantees for general distributions, has been proposed so far. Here, we provide the first such algorithm and guarantees.

The passive nearest-neighbor classifier is popular among theorists and practitioners alike [Fix and Hodges, 1989, Cover and Hart, 1967, Stone, 1977, Kulkarni and Posner, 1995]. This paradigm is applicable in general metric spaces, and its simplicity is an attractive feature for both implementation and analysis. When appropriately regularized [e.g. Stone, 1977, Devroye and Györfi, 1985, von Luxburg and Bousquet, 2004, Gottlieb et al., 2010, Kontorovich and Weiss, 2015] this type of learner can be made Bayes consistent. Another desirable property of nearest-neighbor-based methods is their ability to generalize at a rate that scales with the intrinsic data dimension, which can be much lower than that of the ambient space [Kpotufe, 2011, Gottlieb et al., 2014a, 2016a, Chaudhuri and Dasgupta, 2014]. Furthermore, margin-based regularization makes nearest neighbors ideally suited for sample compression, which yields a compact representation, faster classification runtime, and improved generalization performance [Gottlieb et al., 2014b, Kontorovich and Weiss, 2015]. The resulting error guarantees can be stated in terms of the sample's noisy-margin, which depends on the distances between differently-labeled examples in the input sample.

**Our contribution**. We propose MARMANN, a non-parametric pool-based active learning algorithm that obtains an error guarantee competitive with that of a noisy-margin-based passive learner, but can provably use significantly fewer labels. This is the first non-parametric active learner for general metric spaces that achieves prediction error that is competitive with passive learning for general distributions, and provably improves label complexity.

**Our approach**. Previous passive learning approaches to classification using nearest-neighbor rules under noisy-margin assumptions [Gottlieb et al., 2014b, 2016b] provide statistical guarantees using sample compression bounds [Graepel et al., 2005]. The finite-sample guarantees depend on the number of noisy labels relative to an optimal margin scale. A central challenge in the active setting is performing model selection (selecting the margin scale) with a low label complexity. A key insight that we exploit in this work is that by designing a new labeling scheme for the compression set, we can construct the compression set and estimate its error with label-efficient procedures. We obtain statistical guarantees for this approach using a generalized sample compression analysis. We derive a label-efficient (as well as computationally efficient) active model-selection procedure. This procedure finds a good scale by estimating the sample error for some scales, using a small number of active querying rounds. Crucially, unlike cross-validation, our model-selection procedure does not require a number of labels that depends on the worst possible scale, nor does it test many scales. This allows our label complexity bounds to be low, and to depend only on the final scale selected by the algorithm. Our error guarantee is a constant factor over the error guarantee of the passive learner of Gottlieb et al. [2016b]. An approach similar to Gottlieb et al. [2016b], proposed in Gottlieb et al. [2014a], has been shown to be Bayes consistent [Kontorovich and Weiss, 2015]. The Bayes-consistency of the passive version of our approach is the subject of ongoing work.

**Paper outline**. We define the setting and notations in Section 2. In Section 3 we provide our main result, Theorem 3.2, giving error and label complexity guarantees for MARMANN. Section 4 shows how to set the nearest neighbor rule for a given scale, and Section 5 describes the model selection procedure. Some of the analysis is omitted due to lack of space. The full analysis is available at Kontorovich et al. [2016].

## 2 Setting and notations

We consider learning in a general metric space $(\mathcal{X}, \rho)$, where $\mathcal{X}$ is a set and $\rho$ is the metric on $\mathcal{X}$. Our problem setting is that of *classification* of the instance space $\mathcal{X}$ into some finite label set $\mathcal{Y}$. Assume that there is some distribution $\mathcal{D}$ over $\mathcal{X} \times \mathcal{Y}$, and let $S \sim \mathcal{D}^m$ be a labeled sample of size $m$, where $m$ is an integer. Denote the sequence of unlabeled points in $S$ by $\mathbb{U}(S)$. We sometimes treat $S$ and $\mathbb{U}(S)$ as multisets, since the order is unimportant. The error of a classifier $h : \mathcal{X} \to \mathcal{Y}$ on $\mathcal{D}$ is denoted $\text{err}(h, \mathcal{D}) := \mathbb{P}[h(X) \neq Y]$, where $(X, Y) \sim \mathcal{D}$. The empirical error on a labeled sample $S$ instantiates to $\text{err}(h, S) = \frac{1}{|S|} \sum \mathbb{I}[h(X) \neq Y]$. A passive learner receives a labeled sample $S_{\text{in}}$ as input. An active learner receives the unlabeled part of the sample $U_{\text{in}} := \mathbb{U}(S_{\text{in}})$ as input, and

is allowed to adaptively select examples from $U_{\mathrm{in}}$ and request their label from $S_{\mathrm{in}}$. When either learner terminates, it outputs a classifier $\hat{h} : \mathcal{X} \to \mathcal{Y}$, with the goal of achieving a low $\mathrm{err}(\hat{h}, \mathcal{D})$. An additional goal of the active learner is to achieve a performance competitive with that of the passive learner, while querying considerably fewer labels.

The diameter of a set $A \subseteq \mathcal{X}$ is defined by $\mathsf{diam}(A) := \sup_{a,a' \in A} \rho(a, a')$. Denote the index of the closest point in $U$ to $x \in \mathcal{X}$ by $\kappa(x, U) := \mathrm{argmin}_{i:x_i \in U} \rho(x, x_i)$. We assume here and throughout this work that when there is more than one minimizer for $\rho(x, x_i)$, ties are broken arbitrarily (but in a consistent fashion). For a set $Z \subseteq \mathcal{X}$, denote $\kappa(Z, U) := \{\kappa(z, U) \mid z \in Z\}$. Any labeled sample $S = ((x_i, y_i))_{i \in [k]}$ naturally induces the nearest-neighbor classifier $h_S^{\mathrm{nn}} : \mathcal{X} \to \mathcal{Y}$, via $h_S^{\mathrm{nn}}(x) := y_{\kappa(x, \mathbb{U}(S))}$.

For $x \in \mathcal{X}$, and $t > 0$, denote by $\mathsf{ball}(x, t)$ the (closed) ball of radius $t$ around $x$: $\mathsf{ball}(x, t) := \{x' \in \mathcal{X} \mid \rho(x, x') \leq t\}$. The *doubling dimension*, the effective dimension of the metric space, which controls generalization and runtime performance of nearest-neighbors [Kpotufe, 2011, Gottlieb et al., 2014a], is defined as follows. Let $\lambda = \lambda(\mathcal{X})$ be the smallest number such that every ball in $\mathcal{X}$ can be covered by $\lambda$ balls of half its radius, where all balls are centered at points of $\mathcal{X}$. Formally, $\lambda(\mathcal{X}) := \min\{\lambda \in \mathbb{N} : \forall x \in \mathcal{X}, r > 0, \quad \exists x_1, \ldots, x_\lambda \in \mathcal{X} : \mathsf{ball}(x, r) \subseteq \cup_{i=1}^{\lambda} \mathsf{ball}(x_i, r/2)\}$. Then the doubling dimension of $\mathcal{X}$ is defined by $\mathsf{ddim}(\mathcal{X}) := \log_2 \lambda$. In line with modern literature, we work in the low-dimension, big-sample regime, where the doubling dimension is assumed to be constant and hence sample complexity and algorithmic runtime may depend on it exponentially. This exponential dependence is unavoidable, even under margin assumptions, as previous analysis [Kpotufe, 2011, Gottlieb et al., 2014a] indicates.

A set $A \subseteq \mathcal{X}$ is $t$-*separated* if $\inf_{a,a' \in A: a \neq a'} \rho(a, a') \geq t$. For $A \subseteq B \subseteq \mathcal{X}$, the set $A$ is a $t$-*net* of $B$ if $A$ is $t$-separated and $B \subseteq \bigcup_{a \in A} \mathsf{ball}(a, t)$. Constructing a minimum size $t$-net for a general set $B$ is NP-hard [Gottlieb and Krauthgamer, 2010], however efficient procedures exist for constructing some $t$-net [Krauthgamer and Lee, 2004, Gottlieb et al., 2014b]. The size of any $t$-net is at most $2^{\mathsf{ddim}(B)}$ times the smallest possible size (see Kontorovich et al. [2016]). In addition, the size of any $t$-net is at most $\lceil \mathsf{diam}(B)/t \rceil^{\mathsf{ddim}(\mathcal{X})+1}$ [Krauthgamer and Lee, 2004]. Throughout the paper, we fix a deterministic procedure for constructing a $t$-net, and denote its output for a multiset $U \subseteq \mathcal{X}$ by $\mathsf{Net}(U, t)$. Let $\mathsf{Par}(U, t)$ be a partition of $\mathcal{X}$ into regions induced by $\mathsf{Net}(U, t)$, that is: for $\mathsf{Net}(U, t) = \{x_1, \ldots, x_N\}$, define $\mathsf{Par}(U, t) := \{P_1, \ldots, P_N\}$, where $P_i = \{x \in \mathcal{X} \mid \kappa(x, \mathsf{Net}(U, t)) = i\}$. For $t > 0$, denote $\mathcal{N}(t) := |\mathsf{Net}(U_{\mathrm{in}}, t)|$. For a labeled multiset $S \subseteq \mathcal{X} \times \mathcal{Y}$ and $y \in \mathcal{Y}$, denote $S^y := \{x \mid (x, y) \in S\}$; in particular, $\mathbb{U}(S) = \cup_{y \in \mathcal{Y}} S^y$.

## 3   Main results

Non-parameteric binary classification admits performance guarantees that scale with the sample's noisy-margin [von Luxburg and Bousquet, 2004, Gottlieb et al., 2010, 2016b]. We say that a labeled multiset $S$ is $(\nu, t)$-*separated*, for $\nu \in [0, 1]$ and $t > 0$ (representing a margin $t$ with noise $\nu$), if one can remove a $\nu$-fraction of the points in $S$, and in the resulting multiset, points with different labels are at least $t$-far from each other. Formally, $S$ is $(\nu, t)$-separated if there exists a subsample $\tilde{S} \subseteq S$ such that $|S \setminus \tilde{S}| \leq \nu|S|$ and $\forall y_1 \neq y_2 \in \mathcal{Y}, a \in \tilde{S}^{y_1}, b \in \tilde{S}^{y_2}$, we have $\rho(a, b) \geq t$. For a given labeled sample $S$, denote by $\nu(t)$ the smallest value $\nu$ such that $S$ is $(\nu, t)$-separated. Gottlieb et al. [2016b] propose a passive learner with the following guarantees as a function of the separation of $S$. Setting $\alpha := m/(m - N)$, define the following form of a generalization bound:

$$\mathsf{GB}(\epsilon, N, \delta, m, k) := \alpha\epsilon + \frac{2}{3} \frac{(N+1)\log(mk) + \log(\frac{1}{\delta})}{m - N} + \frac{3}{\sqrt{2}} \sqrt{\frac{\alpha\epsilon((N+1)\log(mk) + \log(\frac{1}{\delta}))}{m - N}}.$$

**Theorem 3.1** (Gottlieb et al. [2016b]). *Let $m$ be an integer, $\mathcal{Y} = \{0, 1\}$, $\delta \in (0, 1)$. There exists a passive learning algorithm that returns a nearest-neighbor classifier $h_{S_{\mathrm{pas}}}^{\mathrm{nn}}$, where $S_{\mathrm{pas}} \subseteq S_{\mathrm{in}}$, such that, with probability $1 - \delta$,*

$$\mathrm{err}(h_{S_{\mathrm{pas}}}^{\mathrm{nn}}, \mathcal{D}) \leq \min_{t > 0 : \mathcal{N}(t) < m} \mathsf{GB}(\nu(t), \mathcal{N}(t), \delta, m, 1).$$

The passive algorithm of Gottlieb et al. [2016b] generates $S_{\mathrm{pas}}$ of size approximately $\mathcal{N}(t)$ for the optimal scale $t > 0$ (found by searching over all scales), removing the $|S_{\mathrm{in}}|\nu(t)$ points that

obstruct the $t$-separation between different labels in $S_{\mathrm{in}}$, and then selecting a subset of the remaining labeled examples to form $S_{\mathrm{pas}}$, so that the examples are a $t$-net for $S_{\mathrm{in}}$. We propose a different approach for generating a compression set for a nearest-neighbor rule. This approach, detailed in the following sections, does not require finding and removing all the obstructing points in $S_{\mathrm{in}}$, and can be implemented in an active setting using a small number of labels. The resulting active learning algorithm, MARMANN, has an error guarantee competitive with that of the passive learner and a label complexity that can be significantly lower. Our main result is the following guarantee for MARMANN.

**Theorem 3.2.** *Let* $S_{\mathrm{in}} \sim \mathcal{D}^m$, *where* $m \geq \max(6, |\mathcal{Y}|)$, $\delta \in (0, \frac{1}{4})$. *Let* $\hat{S}$ *be the output of* MARMANN$(U_{\mathrm{in}}, \delta)$, *where* $\hat{S} \subseteq \mathcal{X} \times \mathcal{Y}$, *and let* $\hat{N} := |\hat{S}|$. *Let* $\hat{h} := h_{\hat{S}}^{\mathrm{nn}}$ *and* $\hat{\epsilon} := \mathrm{err}(\hat{h}, S_{\mathrm{in}})$, *and denote* $\hat{G} := \mathrm{GB}(\hat{\epsilon}, \hat{N}, \delta, m, 1)$. *With a probability of* $1 - \delta$ *over* $S_{\mathrm{in}}$ *and randomness of* MARMANN,

$$\mathrm{err}(\hat{h}, \mathcal{D}) \leq 2\hat{G} \leq O \left( \min_{t > 0 : \mathcal{N}(t) < m} \mathrm{GB}(\nu(t), \mathcal{N}(t), \delta, m, 1) \right),$$

*and the number of labels from* $S_{\mathrm{in}}$ *requested by* MARMANN *is at most*

$$O \left( \log^3(\frac{m}{\delta}) \left( \frac{1}{\hat{G}} \log(\frac{1}{\hat{G}}) + m\hat{G} \right) \right).$$

*Here* $O(\cdot)$ *hides only universal numerical constants.*

To observe the advantages of MARMANN over a passive learner, consider a scenario in which the upper bound GB of Theorem 3.1, as well as the Bayes error of $\mathcal{D}$, are of order $\Theta(1/\sqrt{m})$. Then $\hat{G} = \Theta(1/\sqrt{m})$ as well. Therefore, MARMANN obtains a prediction error guarantee of $\Theta(1/\sqrt{m})$, similarly to the passive learner, but it uses only $\tilde{\Theta}(\sqrt{m})$ labels instead of $m$. Moreover, no learner that selects labels randomly from $S_{\mathrm{in}}$ can compete with MARMANN: In Kontorovich et al. [2016] we adapt an argument of Devroye et al. [1996] to show that for any passive learner that uses $\tilde{\Theta}(\sqrt{m})$ random labels from $S_{\mathrm{in}}$, there exists a distribution $\mathcal{D}$ with the above properties, for which the prediction error of the passive learner in this case is $\tilde{\Omega}(m^{-1/4})$, a decay rate which is almost quadratically slower than the $O(1/\sqrt{m})$ rate achieved by MARMANN. Thus, the guarantees of MARMANN cannot be matched by any passive learner.

MARMANN operates as follows. First, a scale $\hat{t} > 0$ is selected, by calling $\hat{t} \leftarrow$ SelectScale$(\delta)$, where SelectScale is our model selection procedure. SelectScale has access to $U_{\mathrm{in}}$, and queries labels from $S_{\mathrm{in}}$ as necessary. It estimates the generalization error bound GB for several different scales, and executes a procedure similar to binary search to identify a good scale. The binary search keeps the number of estimations (and thus requested labels) small. Crucially, our estimation procedure is designed to prevent the search from spending a number of labels that depends on the net size of the smallest possible scale $t$, so that the total label complexity of MARMANN depends only on error of the selected $\hat{t}$. Second, the selected scale $\hat{t}$ is used to generate the compression set by calling $\hat{S} \leftarrow$ GenerateNNSet$(\hat{t}, [\mathcal{N}(\hat{t})], \delta)$, where GenerateNNSet is our compression set generation procedure. For clarity of presentation, we first introduce in Section 4 the procedure GenerateNNSet, which determines the compression set for a given scale, and then in Section 5, we describe how SelectScale chooses the appropriate scale.

## 4 Active nearest-neighbor at a given scale

The passive learner of Gottlieb et al. [2014a, 2016b] generates a compression set by first finding and removing from $S_{\mathrm{in}}$ all points that obstruct $(\nu, t)$-separation at a given scale $t > 0$. We propose below a different approach for generating a compression set, which seems more conducive to active learning: as we show below, it also generates a low-error nearest neighbor rule, just like the passive approach. At the same time, it allows us to estimate the error on many different scales using few label queries. A small technical difference, which will be evident below, is that in this new approach, examples in the compression set might have a different label than their original label in $S_{\mathrm{in}}$. Standard sample compression analysis [e.g. Graepel et al., 2005] assumes that the classifier is determined by a small number of labeled examples from $S_{\mathrm{in}}$. This does not allow the examples in the compression set to have a different label than their original label in $S_{\mathrm{in}}$. Therefore, we require a slight generalization of previous compression analysis, which allows setting arbitrary labels for examples that are assigned to the compression set. The following theorem quantifies the effect of this change on generalization.

**Theorem 4.1.** *Let $m \geq |\mathcal{Y}|$ be an integer, $\delta \in (0, \frac{1}{4})$. Let $S_{\mathrm{in}} \sim \mathcal{D}^m$. With probability at least $1 - \delta$, if there exist $N < m$ and $S \subseteq (\mathcal{X} \times \mathcal{Y})^N$ such that $\mathbb{U}(S) \subseteq U_{\mathrm{in}}$ and $\epsilon := \mathrm{err}(h_S^{\mathrm{nn}}, S_{\mathrm{in}}) \leq \frac{1}{2}$, then $\mathrm{err}(h_S^{\mathrm{nn}}, \mathcal{D}) \leq \mathrm{GB}(\epsilon, N, \delta, m, |\mathcal{Y}|) \leq 2\mathrm{GB}(\epsilon, N, 2\delta, m, 1)$.*

The proof is similar to that of standard sample compression schemes. If the compression set includes only the original labels, the compression analysis of Gottlieb et al. [2016b] gives the bound $\mathrm{GB}(\epsilon, N, \delta, m, 1)$. Thus the effect of allowing the labels to change is only logarithmic in $|\mathcal{Y}|$, and does not appreciably degrade the prediction error.

We now describe the generation of the compression set for a given scale $t > 0$. Recall that $\nu(t)$ is the smallest value for which $S_{\mathrm{in}}$ is $(\nu, t)$-separated. We define two compression sets. The first one, denoted $S_{\mathrm{a}}(t)$, represents an ideal compression set, which induces an empirical error of at most $\nu(t)$, but calculating it might require many labels. The second compression set, denoted $\hat{S}_{\mathrm{a}}(t)$, represents an approximation to $S_{\mathrm{a}}(t)$, which can be constructed using a small number of labels, and induces a sample error of at most $4\nu(t)$ with high probability. MARMANN constructs only $\hat{S}_{\mathrm{a}}(t)$, while $S_{\mathrm{a}}(t)$ is defined for the sake of analysis only.

We first define the ideal set $S_{\mathrm{a}}(t) := \{(x_1, y_1), \ldots, (x_N, y_N)\}$. The examples in $S_{\mathrm{a}}(t)$ are the points in $\mathsf{Net}(U_{\mathrm{in}}, t/2)$, and the label of each example is the majority label out of the examples in $S_{\mathrm{in}}$ to which $x_i$ is closest. Formally, $\{x_1, \ldots, x_N\} := \mathsf{Net}(U_{\mathrm{in}}, t/2)$, and for $i \in [N]$, $y_i := \mathrm{argmax}_{y \in \mathcal{Y}} |S^y \cap P_i|$, where $P_i = \{x \in \mathcal{X} \mid \kappa(x, \mathsf{Net}(U, t/2)) = i\} \in \mathsf{Par}(U_{\mathrm{in}}, t/2)$. For $i \in [N]$, let $\Lambda_i := \tilde{S}^{y_i} \cap P_i$. The following lemma bounds the empirical error of $h_{S_{\mathrm{a}}(t)}^{\mathrm{nn}}$.

**Lemma 4.2.** *For every $t > 0$, $\mathrm{err}(h_{S_{\mathrm{a}}(t)}^{\mathrm{nn}}, S_{\mathrm{in}}) \leq \nu(t)$.*

*Proof.* Since $\mathsf{Net}(U_{\mathrm{in}}, t/2)$ is a $t/2$-net, $\mathrm{diam}(P) \leq t$ for any $P \in \mathsf{Par}(U_{\mathrm{in}}, t/2)$. Let $\tilde{S} \subseteq S$ be a subsample that witnesses the $(\nu(t), t)$-separation of $S$, so that $|\tilde{S}| \geq m(1 - \nu(t))$, and for any two points $(x, y), (x', y') \in \tilde{S}$, if $\rho(x, x') \leq t$ then $y = y'$. Denote $\tilde{U} := \mathbb{U}(\tilde{S})$. Since $\max_{P \in \mathsf{Par}(U_{\mathrm{in}}, t/2)} \mathrm{diam}(P) \leq t$, for any $i \in [N]$ all the points in $\tilde{U} \cap P_i$ must have the same label in $\tilde{S}$. Therefore, $\exists y \in \mathcal{Y}$ such that $\tilde{U} \cap P_i \subseteq \tilde{S}^y \cap P_i$. Hence $|\tilde{U} \cap P_i| \leq |\Lambda_i|$. It follows

$$m \cdot \mathrm{err}(h_{S_{\mathrm{a}}(t)}^{\mathrm{nn}}, S_{\mathrm{in}}) \leq |S| - \sum_{i \in [N]} |\Lambda_i| \leq |S| - \sum_{i \in [N]} |\tilde{U} \cap P_i| = |S| - |\tilde{S}| = m \cdot \nu(t).$$

Dividing by $m$ we get the statement of the theorem. $\qquad\square$

Now, calculating $S_{\mathrm{a}}(t)$ requires knowing most of the labels in $S_{\mathrm{in}}$. MARMANN constructs instead an approximation $\hat{S}_{\mathrm{a}}(t)$, in which the examples are the points in $\mathsf{Net}(U_{\mathrm{in}}, t/2)$ (so that $\mathbb{U}(\hat{S}_{\mathrm{a}}(t)) = \mathbb{U}(S_{\mathrm{a}}(t))$ ), but the labels are determined using a bounded number of labels requested from $S_{\mathrm{in}}$. The labels in $\hat{S}_{\mathrm{a}}(t)$ are calculated by the simple procedure GenerateNNSet given in Alg. 1. The empirical error of the output of GenerateNNSet is bounded in Theorem 4.3 below.[1]

A technicality in Alg. 1 requires explanation: In MARMANN, the generation of $\hat{S}_{\mathrm{a}}(t)$ will be split into several calls to GenerateNNSet, so that different calls determine the labels of different points in $\hat{S}_{\mathrm{a}}(t)$. Therefore GenerateNNSet has an additional argument $I$, which specifies the indices of the points in $\mathsf{Net}(U_{\mathrm{in}}, t/2)$ for which the labels should be returned this time. Crucially, if during the run of MARMANN, GenerateNNSet is called again for the same scale $t$ and the same point in $\mathsf{Net}(U_{\mathrm{in}}, t/2)$, then GenerateNNSet returns the same label that it returned before, rather than recalculating it using fresh labels from $S_{\mathrm{in}}$. This guarantees that despite the randomness in GenerateNNSet, the full $\hat{S}_{\mathrm{a}}(t)$ is well-defined within any single run of MARMANN, and is distributed like the output of GenerateNNSet$(t, [\mathcal{N}(t/2)], \delta)$, which is convenient for the analysis.

**Theorem 4.3.** *Let $\hat{S}_{\mathrm{a}}(t)$ be the output of GenerateNNSet$(t, [\mathcal{N}(t/2)], \delta)$. With a probability at least $1 - \frac{\delta}{2m^2}$, we have $\mathrm{err}(h_S^{\mathrm{nn}}, S_{\mathrm{in}}) \leq 4\nu(t)$. Denote this event by $E(t)$.*

**Algorithm 1** GenerateNNSet$(t, I, \delta)$

---

**input** Scale $t > 0$, a target set $I \subseteq [\mathcal{N}(t/2)]$, confidence $\delta \in (0,1)$.
**output** A labeled set $S \subseteq \mathcal{X} \times \mathcal{Y}$ of size $|I|$
    $\{x_1, \ldots, x_N\} \leftarrow \mathsf{Net}(U_{\mathrm{in}}, t/2)$, $\{P_1, \ldots, P_N\} \leftarrow \mathsf{Par}(U_{\mathrm{in}}, t/2)$, $S \leftarrow ()$
    **for** $i \in I$ **do**
        **if** $\hat{y}_i$ has not already been calculated for $U_{\mathrm{in}}$ with this values of $t$ **then**
            Draw $Q := \lceil 18 \log(2m^3/\delta) \rceil$ points uniformly at random from $P_i$ and query their labels.
            Let $\hat{y}_i$ be the majority label observed in these $Q$ queries.
        **end if**
        $S \leftarrow S \cup \{(x_i, \hat{y}_i)\}$.
    **end for**
    Output $S$

---

*Proof.* By Lemma 4.2, $\mathrm{err}(h^{\mathrm{nn}}_{S_{\mathrm{a}}(t)}, S_{\mathrm{in}}) \leq \nu(t)$. In $S_{\mathrm{a}}(t)$, the labels assigned to each point in $\mathsf{Net}(U_{\mathrm{in}}, t/2)$ are the majority labels (based on $S_{\mathrm{in}}$) of the points in the regions in $\mathsf{Par}(U_{\mathrm{in}}, t/2)$. Denote the majority label for region $P_i$ by $y_i := \mathrm{argmax}_{y \in \mathcal{Y}} |S^y \cap P_i|$. We now compare these labels to the labels $\hat{y}_i$ assigned by Alg. 1. Let $p(i) = |\Lambda_i|/|P_i|$ be the fraction of points in $P_i$ which are labeled by the majority label $y_i$. Let $\hat{p}(i)$ be the fraction of labels equal to $y_i$ out of those queried by Alg. 1 in round $i$. Let $\beta := 1/6$. By Hoeffding's inequality and union bounds, we have that with a probability of at least $1 - \mathcal{N}(t/2)\exp(-\frac{Q}{18}) \geq 1 - \frac{\delta}{2m^2}$, we have $\max_{i \in [\mathcal{N}(t/2)]} |\hat{p}(i) - p(i)| \leq \beta$. Denote this "good" event by $E'$. We now prove that $E' \Rightarrow E(t)$. Let $J \subseteq [\mathcal{N}(t/2)] = \{i \mid \hat{p}(i) > \frac{1}{2}\}$. It can be easily seen that $\hat{y}_i = y_i$ for all $i \in J$. Therefore, for all $x$ such that $\kappa(x, \mathbb{U}(S_{\mathrm{a}}(t))) \in J$, $h^{\mathrm{nn}}_S(x) = h^{\mathrm{nn}}_{S_{\mathrm{a}}(t)}(x)$, and hence $\mathrm{err}(h^{\mathrm{nn}}_S, U_{\mathrm{in}}) \leq \mathbb{P}_{X \sim U_{\mathrm{in}}}[\kappa(X, \mathbb{U}(S_{\mathrm{a}}(t))) \notin J] + \mathrm{err}(h^{\mathrm{nn}}_{S_{\mathrm{a}}(t)}, U_{\mathrm{in}})$. The second term is at most $\nu(t)$, and it remains to bound the first term, on the condition that $E'$ holds. We have $\mathbb{P}_{X \sim U}[\kappa(X, \mathbb{U}(S_{\mathrm{a}}(t))) \notin J] = \frac{1}{m} \sum_{i \notin J} |P_i|$. If $E'$ holds, then for any $i \notin J$, $p(i) \leq \frac{1}{2} + \beta$, therefore $|P_i| - |\Lambda_i| = (1 - p(i))|P_i| \geq (\frac{1}{2} - \beta)|P_i|$. Therefore

$$1 - \frac{1}{m} \sum_{i \notin J} |\Lambda_i| \geq \frac{1}{m} \sum_{i \notin J} |P_i|(\frac{1}{2} - \beta) = \mathbb{P}_{X \sim U}[\kappa(X, \mathbb{U}(S_{\mathrm{a}}(t))) \notin J](\frac{1}{2} - \beta).$$

On the other hand, as in the proof of Lemma 4.2, $1 - \frac{1}{m} \sum_{i \in [\mathcal{N}(t/2)]} |\Lambda_i| \leq \nu(t)$. Thus, under $E'$, $\mathbb{P}_{X \sim U}[\kappa(X, S) \notin J] \leq \frac{\nu(t)}{\frac{1}{2} - \beta} = 3\nu(t)$. It follows that under $E'$, $\mathrm{err}(h^{\mathrm{nn}}_S, U_{\mathrm{in}}) \leq 4\nu(t)$. $\qquad\square$

## 5 Model Selection

We now show how to select the scale $\hat{t}$ that will be used to generate the output nearest-neighbor rule. The main challenge is to do this with a low label complexity: Generating the full classification rule for scale $t$ requires a number of labels that depends on $\mathcal{N}(t)$, which might be very large. We would like the label complexity of MARMANN to depend only on $\mathcal{N}(\hat{t})$ (where $\hat{t}$ is the selected scale), which is of the order $m\hat{G}$. Therefore, during model selection we can only invest a bounded number of labels in each tested scale. In addition, to keep the label complexity low, we cannot test all scales.

For $t > 0$, let $\hat{S}_{\mathrm{a}}(t)$ be the model that MARMANN would generate if the selected scale were set to $t$. Our model selection procedure performs a search, similar to binary search, over the possible scales. For each tested scale $t$, the procedure estimates $\epsilon(t) := \mathrm{err}(h^{\mathrm{nn}}_{\hat{S}_{\mathrm{a}}(t)}, S)$ within a certain accuracy, using an estimation procedure we call EstimateErr. EstimateErr outputs an estimate $\hat{\epsilon}(t)$ of $\epsilon(t)$, up to a given accuracy $\theta > 0$, using labels requested from $S_{\mathrm{in}}$. It draws random examples from $S_{\mathrm{in}}$, asks for their label, and calls GenerateNNSet (which also might request labels) to find the prediction error of $h^{\mathrm{nn}}_{\hat{S}_{\mathrm{a}}(t)}$ on these random examples. The estimate $\hat{\epsilon}(t)$ is set to this prediction error. The number of random examples drawn by EstimateErr is determined based on the accuracy $\theta$, using empirical Bernstein bounds [Maurer and Pontil, 2009]. Theorem 5.1 gives a guarantee for the accuracy and label complexity of EstimateErr. The full implementation of EstimateErr and the proof of Theorem 5.1 can be found in the long version of this paper Kontorovich et al. [2016].

**Theorem 5.1.** *Let $t, \theta > 0$ and $\delta \in (0, 1)$, and let $\hat{\epsilon}(t) \leftarrow$ EstimateErr$(t, \theta, \delta)$. Let $Q$ be as defined in Alg. 1. The following properties (which we denote below by $V(t)$) hold with a probability of $1 - \frac{\delta}{2m^2}$ over the randomness of EstimateErr (and conditioned on $\hat{S}_a(t)$).*

1. *If $\hat{\epsilon}(t) \leq \theta$, then $\epsilon(t) \leq 5\theta/4$. Otherwise, $\frac{4\epsilon(t)}{5} \leq \hat{\epsilon}(t) \leq \frac{4\epsilon(t)}{3}$.*

2. EstimateErr *requests at most $\frac{520(Q+1)\log(\frac{1040m^2}{\delta\psi'})}{\psi'}$ labels, where $\psi' := \max(\theta, \epsilon(t))$.*

The model selection procedure SelectScale, given in Alg. 2, implements its search based on the guarantees in Theorem 5.1. First, we introduce some notation. Let $G^* = \min_t \text{GB}(\nu(t), \mathcal{N}(t), \delta, m, 1)$. We would like MARMANN to obtain a generalization guarantee that is competitive with $G^*$. Denote $\phi(t) := ((\mathcal{N}(t) + 1)\log(m) + \log(\frac{1}{\delta}))/m$, and let $G(\epsilon, t) := \epsilon + \frac{2}{3}\phi(t) + \frac{3}{\sqrt{2}}\sqrt{\epsilon\phi(t)}$. Note that for all $\epsilon, t$,

$$\text{GB}(\epsilon, \mathcal{N}(t), \delta, m, 1) = \frac{m}{m - \mathcal{N}(t)}G(\epsilon, t).$$

When referring to $G(\nu(t), t)$, $G(\epsilon(t), t)$, or $G(\hat{\epsilon}(t), t)$ we omit the second $t$ for brevity. Instead of directly optimizing GB, we will select a scale based on our estimate $G(\hat{\epsilon}(t))$ of $G(\epsilon(t))$.

Let Dist denote the set of pairwise distances in the unlabeled dataset $U_{\text{in}}$ (note that $|\text{Dist}| < \binom{m}{2}$). We remove from Dist some distances, so that the remaining distances have a net size $\mathcal{N}(t)$ that is monotone non-increasing in $t$. We also remove values with a very large net size. Concretely, define

$$\text{Dist}_{\text{mon}} := \text{Dist} \setminus \{t \mid \mathcal{N}(t) + 1 > m/2\} \setminus \{t \mid \exists t' \in \text{Dist}, t' < t \text{ and } \mathcal{N}(t') < \mathcal{N}(t)\}.$$

Then for all $t, t' \in \text{Dist}_{\text{mon}}$ such that $t' < t$, we have $\mathcal{N}(t') \geq \mathcal{N}(t)$. The output of SelectScale is always a value in $\text{Dist}_{\text{mon}}$. The following lemma shows that it suffices to consider these scales.

**Lemma 5.2.** *Assume $m \geq 6$ and let $t_m^* \in \text{argmin}_{t \in \text{Dist}} G(\nu(t))$. If $G^* \leq 1/3$ then $t_m^* \in \text{Dist}_{\text{mon}}$.*

*Proof.* Assume by way of contradiction that $t_m^* \in \text{Dist} \setminus \text{Dist}_{\text{mon}}$. First, since $G(\nu(t_m^*)) \leq G^* \leq 1/3$ we have $\frac{\mathcal{N}(t_m^*)+1}{m-\mathcal{N}(t_m^*)}\log(m) \leq \frac{1}{2}$. Therefore, since $m \geq 6$, it is easy to verify $\mathcal{N}(t_m^*) + 1 \leq m/2$. Therefore, by definition of $\text{Dist}_{\text{mon}}$ there exists a $t \leq t_m^*$ with $\phi(t) < \phi(t_m^*)$. Since $\nu(t)$ is monotone over all of $t \in \text{Dist}$, we also have $\nu(t) \leq \nu(t_m^*)$. Now, $\phi(t) < \phi(t_m^*)$ and $\nu(t) \leq \nu(t_m^*)$ together imply that $G(\nu(t)) < G(\nu(t_m^*))$, a contradiction. Hence, $t_m^* \in \text{Dist}_{\text{mon}}$. $\square$

SelectScale follows a search similar to binary search, however the conditions for going right and for going left are not complementary. The search ends when either none of these two conditions hold, or when there is nothing left to try. The final output of the algorithm is based on minimizing $G(\hat{\epsilon}(t))$ over some of the values tested during search.

For $c > 0$, define $\gamma(c) := 1 + \frac{2}{3c} + \frac{3}{\sqrt{2c}}$ and $\tilde{\gamma}(c) := \frac{1}{c} + \frac{2}{3} + \frac{3}{\sqrt{2c}}$. For all $t, \epsilon > 0$ we have the implications

$$\epsilon \geq c\phi(t) \ \Rightarrow \ \gamma(c)\epsilon \geq G(\epsilon, t) \quad \text{and} \quad \phi(t) \geq c\epsilon \ \Rightarrow \ \tilde{\gamma}(c)\phi(t) \geq G(\epsilon, t). \tag{1}$$

The following lemma uses Eq. (1) to show that the estimate $G(\hat{\epsilon}(t))$ is close to the true $G(\epsilon(t))$.

**Lemma 5.3.** *Let $t > 0$, $\delta \in (0, 1)$, and suppose that SelectScale calls $\hat{\epsilon}(t) \leftarrow$ EstimateErr$(t, \phi(t), \delta)$. Suppose that $V(t)$ as defined in Theorem 5.1 holds. Then $\frac{1}{6}G(\hat{\epsilon}(t)) \leq G(\epsilon(t)) \leq 6.5G(\hat{\epsilon}(t))$.*

*Proof.* Under $V(t)$, we have that if $\hat{\epsilon}(t) < \phi(t)$ then $\epsilon(t) \leq \frac{5}{4}\phi(t)$. In this case, $G(\epsilon(t)) \leq \tilde{\gamma}(4/5)\phi(t) \leq 4.3\phi(t)$, by Eq. (1). Therefore $G(\epsilon(t)) \leq \frac{3 \cdot 4.3}{2}G(\hat{\epsilon}(t))$. In addition, $G(\epsilon(t)) \geq \frac{2}{3}\phi(t)$ (from the definition of $G$), and by Eq. (1) and $\tilde{\gamma}(1) \leq 4$, $\phi(t) \geq \frac{1}{4}G(\hat{\epsilon}(t))$. Therefore $G(\epsilon(t)) \geq \frac{1}{6}G(\hat{\epsilon}(t))$. On the other hand, if $\hat{\epsilon}(t) \geq \phi(t)$, then by Theorem 5.1 $\frac{4}{5}\epsilon(t) \leq \hat{\epsilon}(t) \leq \frac{4}{3}\epsilon(t)$. Therefore $G(\hat{\epsilon}(t)) \leq \frac{4}{3}G(\epsilon(t))$ and $G(\epsilon(t)) \leq \frac{5}{4}G(\hat{\epsilon}(t))$. Taking the worst-case of both possibilities, we get the bounds in the lemma. $\square$

The next theorem bounds the label complexity of SelectScale. Let $\mathcal{T}_{\text{test}} \subseteq \text{Dist}_{\text{mon}}$ be the set of scales that are tested during SelectScale (that is, their $\hat{\epsilon}(t)$ was estimated).

---
**Algorithm 2** SelectScale($\delta$)
---
**input** $\delta \in (0,1)$
**output** Scale $\hat{t}$
   $\mathcal{T} \leftarrow \mathrm{Dist}_{\mathrm{mon}}$,      # $\mathcal{T}$ maintains the current set of possible scales
   **while** $\mathcal{T} \neq \emptyset$ **do**
      $t \leftarrow$ the median value in $\mathcal{T}$      # break ties arbitrarily
      $\hat{\epsilon}(t) \leftarrow \mathsf{EstimateErr}(t, \phi(t), \delta)$.
      **if** $\hat{\epsilon}(t) < \phi(t)$ **then**
         $\mathcal{T} \leftarrow \mathcal{T} \setminus [0, t]$ # go right in the binary search
      **else if** $\hat{\epsilon}(t) > \frac{11}{10}\phi(t)$ **then**
         $\mathcal{T} \leftarrow \mathcal{T} \setminus [t, \infty)$ # go left in the binary search
      **else**
         $t_0 \leftarrow t, \mathcal{T}_0 \leftarrow \{t_0\}$.
         **break** from loop
      **end if**
   **end while**
   **if** $\mathcal{T}_0$ was not set yet **then**
      If the algorithm ever went to the right, let $t_0$ be the last value for which this happened, and let $\mathcal{T}_0 := \{t_0\}$. Otherwise, $\mathcal{T}_0 := \emptyset$.
   **end if**
   Let $\mathcal{T}_L$ be the set of all $t$ that were tested and made the search go left
   Output $\hat{t} := \mathrm{argmin}_{t \in \mathcal{T}_L \cup \mathcal{T}_0} G(\hat{\epsilon}(t))$
---

**Theorem 5.4.** *Suppose that the event $V(t)$ defined in Theorem 5.1 holds for all $t \in \mathcal{T}_{\mathrm{test}}$ for the calls $\hat{\epsilon}(t) \leftarrow \mathsf{EstimateErr}(t, \phi(t), \delta)$. If the output of* SelectScale *is $\hat{t}$, then the number of labels requested by* SelectScale *is at most*

$$19240|\mathcal{T}_{\mathrm{test}}|(Q+1)\frac{1}{G(\epsilon(\hat{t}))}\log(\frac{38480m^2}{\delta G(\epsilon(\hat{t}))}),$$

*where $Q$ is as defined in Alg. 1.*

The following theorem provides a competitive error guarantee for the selected scale $\hat{t}$.

**Theorem 5.5.** *Suppose that $V(t)$ and $E(t)$, defined in Theorem 5.1 and Theorem 4.3, hold for all values $t \in \mathcal{T}_{\mathrm{test}}$, and that $G^* \leq 1/3$. Then* SelectScale *outputs $\hat{t} \in \mathrm{Dist}_{\mathrm{mon}}$ such that*

$$\mathrm{GB}(\epsilon(\hat{t}), \mathcal{N}(\hat{t}), \delta, m, 1) \leq O(G^*),$$

*where $O(\cdot)$ hides numerical constants only.*

The idea of the proof is as follows: First, we show (using Lemma 5.3) that it suffices to prove that $G(\nu(t_m^*)) \geq O(G(\hat{\epsilon}(\hat{t})))$ to derive the bound in the theorem. Now, SelectScale ends in one of two cases: either $\mathcal{T}_0$ is set within the loop, or $\mathcal{T} = \emptyset$ and $\mathcal{T}_0$ is set outside the loop. In the first case, neither of the conditions for turning left and turning right holds for $t_0$, so we have $\hat{\epsilon}(t_0) = \Theta(\phi(t_0))$ (where $\Theta$ hides numerical constants). We show that in this case, whether $t_m^* \geq t_0$ or $t_m^* \leq t_0$, $G(\nu(t_m^*)) \geq O(G(\hat{\epsilon}(t_0)))$. In the second case, there exist (except for edge cases, which are also handled) two values $t_0 \in \mathcal{T}_0$ and $t_1 \in \mathcal{T}_L$ such that $t_0$ caused the binary search to go right, and $t_1$ caused it to go left, and also $t_0 \leq t_1$, and $(t_0, t_1) \cap \mathrm{Dist}_{\mathrm{mon}} = \emptyset$. We use these facts to show that for $t_m^* \geq t_1$, $G(\nu(t_m^*)) \geq O(G(\hat{\epsilon}(t_1)))$, and for $t_m^* \leq t_0$, $G(\nu(t_m^*)) \geq O(G(\hat{\epsilon}(t_0)))$. Since $\hat{t}$ minimizes over a set that includes $t_0$ and $t_1$, this gives $G(\nu(t_m^*)) \geq O(G(\hat{\epsilon}(\hat{t})))$ in all cases.

The proof of the main theorem, Theorem 3.2, which gives the guarantee for MARMANN, is almost immediate from Theorem 4.1, Theorem 4.3, Theorem 5.5 and Theorem 5.4.

### Acknowledgements

Sivan Sabato was partially supported by the Israel Science Foundation (grant No. 555/15). Aryeh Kontorovich was partially supported by the Israel Science Foundation (grants No. 1141/12 and 755/15) and a Yahoo Faculty award. We thank Lee-Ad Gottlieb and Dana Ron for helpful discussions.

## Footnotes

[1]In the case of binary labels ($|\mathcal{Y}| = 2$), the problem of estimating $S_{\mathrm{a}}(t)$ can be formulated as a special case of the benign noise setting for parametric active learning, for which tight lower and upper bounds are provided in Hanneke and Yang [2015]. However, our case is both more general (as we allow multiclass labels) and more specific (as we are dealing with a specific hypothesis class). Thus we provide our own procedure and analysis.

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
