[Reviews · NeurIPS 2016]

Reviewer 1

Summary

This work develops an active learning algorithm based on a nearest neighbor classifier, bounds its error rate in terms of a noisy margin type of complexity, and bounds its query complexity by a value that is typically smaller than would be required of passive learning to achieve the same error rate guarantee. For instance, in the special case that the error bound for the analogous passive learner using m labeled samples is O(1/\sqrt{m}), this active learner also achieves O(1/\sqrt{m}) error bound, but with m unlabeled samples and only \tilde{O}(\sqrt{m}) queries. Their key idea is to note that, if we use a net of a given scale as the reference set in the nearest neighbor classifier, then we only need to query a number of points roughly on the order of the size of such a net in order to produce this classifier, supposing we have a method for selecting the appropriate scale without making too many additional queries. The paper proposes such a scale-selection method, which importantly does not need to query points in a net of scale much smaller than the optimal scale used in the error bound.

Qualitative Assessment

My overall assessment is that this is outstanding work, and should definitely be accepted for publication. The reduction in query complexity compared to passive learning in this work is impressively general, in that the improvements require no additional distribution constraints, instead being phrased purely in a data-dependent way. The downside is that this reduction in query complexity comes at the expense of a constant factor increase in the error rate of the classifier. If the Bayes risk is large, say 0.1, then this constant factor increase immediately makes the result trivial. This result is therefore most interesting when the Bayes risk is small. Indeed, since the query bound has a term Gm, where m is the number of unlabeled samples and G is proportional to the bound on the error rate, the result only reflects improvements of active over passive when the achieved error rate is small (which, in turn, is only possible when the Bayes risk itself is small). A few additional comments for the authors: The bounds on query complexity and error rate are themselves very simple, but being data-dependent it is difficult to compare them to the existing results in the active learning literature (which are typically data-independent distribution-dependent bounds). I wonder whether one can state a-priori guarantees that these bounds will be at most some value (with high probability), for some interesting scenarios (i.e., under conditions on the distribution). Is the loss of a constant factor in the error rate guarantee fundamentally necessary for the gains in query complexity in this work? Is there any sense of optimality of this result? For instance, can one claim optimality of the query complexity given that only a constant factor increase to the error bound is allowed? On line 167, "also" is repeated. The bibliography is inconsistent regarding whether first name initials are given or not.

Confidence in this Review

3-Expert (read the paper in detail, know the area, quite certain of my opinion)


Reviewer 2

Summary

The paper presents an algorithm for active learning of NN classifier. The general method follows finding an optimal scale for the data and then compressing the data set such that only part of the labels need to be acquired. The paper presents the algorithm as well as a generalization bound that shows that it is competitive with any passive learner.

Qualitative Assessment

The paper seems very interesting. However the writing can be greatly improved. The algorithm itself is very complicated and designed such that a bound is achievable. This is fine but i would expect some more explanations about the logic of the method, most active learning techniques utilize the fact that different areas of the feature space have different sensitivity for error, i suspect this is not the case for this method. As a result benefit should arise only when data is "over sampled" allowing to sample the data evenly instead of randomly. It is unclear from the text when do this method provide real benefit. Some examples and some empirical results (even on a "toy" dataset) may improve the paper significantly.

Confidence in this Review

1-Less confident (might not have understood significant parts)


Reviewer 3

Summary

This paper propose a pool-based non-parametric active learning algorithm MARMANN for general metric spaces, which outputs a nearest-neighbor classifier. Prediction error competitive with those obtained by previously passive learners and significant lower label complexity than that of passive learner are guaranteed.

Qualitative Assessment

This paper propose a pool-based non-parametric active learning algorithm MARMANN for general metric spaces and outputs a nearest-neighbor classifier. Guarantees over prediction error competitive with those obtained by previously passive learners and significant lower label complexity than that of passive learner are given. As a premise, I must say that I am not an expert in the theoretical active learning field, so I reviewed the paper with the general interest of a researcher in machine learning. My overall impression is positive, the paper is well-structured, the idea seems interesting from a practical point of view. The paper is quite clearly written an I could follow and understand the presented ideas. The problem and the solution is formally defined in a detailed way. To my knowledge, the problem tackled is novel, and so is its results. As I am not so familiar with the literature on theoretical active learning, I cannot judge whether the work is technically sound and correct, and whether the proof technique is new on the current problem. If this is the case, then I think this work is nice. One possible weak point of this paper is that it is lack of experimental analysis. If the proposed active learning method works well, and code is made publicly available, it would have practical impact since reducing label complexity is a valuable process in machine learning tasks.

Confidence in this Review

1-Less confident (might not have understood significant parts)


Reviewer 4

Summary

The paper proposes a new algorithm for active learning of nearest-neighbor classifiers on metric spaces, inspired by a theoretical analysis of its label complexity. In fact, the authors prove that the novel algorithm requires querying a substantially smaller number of labels than the amount available to a passive learner, while the two enjoy a similar generalization bounds.

Qualitative Assessment

I am not qualified to evaluate this work in term of its relevance within the literature. Therefore my judgment is only about the paper content itself. Also, I have only reviewed the proofs contained in the main paper + the one of Lemma A.1. Theorem 3.2 guarantees a significant improvement upon the passive learner characterized by 3.1. I find the example in line2 141-143 about the 1/sqrt(m) order very helpful and I suggest the authors to include it in the introduction as well. All the proofs I have checked seem sound. Some minor comments on the proofs: - Proofs of Lemma A.1: |M_1| \geq k|M_2| should be part of the assumption of the Theorem instead? Also are "proper" t-nets ever defined formally before? - I would avoid using \Lambda_i at all (line 195 and following). Instead, Both Lemma 4.2 and Theorem 4.3 can make use of p(i). - Line 286: is 4.01 correct? From a quick look I don't get the same number - Could you clarify why line 300 is true? Although not being the central point of this paper, I feel that high-level analysis of the time-complexity of MARMANN vs. respective passive learner could be a nice complement. In term of the presentation, I encourage some restructuring in case the paper is accepted. Although this is a fully theoretical work, some parts of the paper should be made easier to read for non-experts of the area and machine learning practitioners. I invite the authors to reflect on what was recommended for FOCS authors’ here http://windowsontheory.org/2014/02/09/advice-for-focs-authors/ More in particular: - enlarge the introduction, mostly the paragraph of "our approach" stressing the part describing the algorithm. - for a short paper like NIPS’, all detailed proofs could go in the Appendix, unless they are believed to give more insights and help understanding of the paper; sketches of demonstrations are fine. - proofs of Theorems 5.2 and all results in Section 5 are very compressed because of space constraints. I suggest to move them in Appendix in order to dedicate more space to them. - a conclusion is missing To my own taste, notation is a heavy at times: - Is the use of S_in vs. S and U_in vs. U necessary? - \alpha in the generalization bounds could be avoided. - S_a and \tilde{S} can be collapsed? - Q is used without a definition in line 225 (the definition should not only be in the algorithm) - to help readability, each algorithm should be self-included, i.e. every (not obvious) function that is used inside it should be declared in the incipit, or at least pointed to its definition in the text. See: Dist_mon, phi(t) and G in Algorithm 2. - the many definitions given on page 7 makes the following proofs hard to read in my opinion Typos: - ‘err’ is missing in Equation between line 201-202. Also, isn’t the last = actually a <= ? - ‘labeld’ line 223 - 'let \hat{S}_a the model that MARMANN' line 241 . the model or the sample/compression? - missing bracket at line 260 in the definition of phi

Confidence in this Review

1-Less confident (might not have understood significant parts)


Reviewer 5

Summary

Nearest neighbor classifiers based on compression sets have been proposed and analyzed in the passive setting, this paper studies the training of a nearest neighbor classifier using a large unlabeled dataset, but many fewer labels. The authors propose a scheme that nearly matches the error rate of the passive learning algorithm (within a constant factor) but achieves a sample complexity, in some cases, that is significantly smaller than the passive algorithm.

Qualitative Assessment

This being my first introduction to compression sets and its use in analyzing nearest neighbor algorithms, I found this paper interesting. However, one thing that was not ever made clear is how min_t nu(t) relates to the Bayes error. Classical asymptotic results say nearest neighbor classifiers can get with a constant factor of the Bayes error, how do the classifiers (passive and active) relate to these results? It is possible that I missed this discussion, but I feel this was the paper's biggest weakness. The problem is that excess risk is not studied but absolute risk, and its not clear that this non-parametric classifier training procedure converges to the best known model in the class at the best possible rate. Or if this best possible classifier is the bayes classifier. The algorithm itself, while requiring a few steps and a lot of notation, is actually quite simple and intuitive once it is all understood. But in a casual reading, the procedure appears quite complicated because the individual steps are introduced one-by-one and not thoroughly enough to get understanding without diving in. For example, the procedure to estimate the error is simply subsampling and applying Hoeffding like bounds, but one needs to inspect the appendix to realize this. The proofs are clearly argued and appear correct. The authors present an example (i.e. if the bayes error is O(1/sqrt(m)) ) that they argue shows the advantage of the adaptive sampling. Given the large constants throughout, I'm very curious as to whether this algorithm performs well in practice. Any empirical results, of any kind, would have greatly benefitted this work. -------- After Rebuttal period ---------- I have seen the author's comments and updated my review.

Confidence in this Review

2-Confident (read it all; understood it all reasonably well)


Reviewer 6

Summary

This paper studies active learning for nearest neighbor classifiers in metric spaces. Built upon a series of work on nearest neighbor classifiers (e.g. Gottlieb and Kontorovich, 2015), the paper provides an active learning algorithm for nearest neighbors, dubbed MARMANN, that has nontrivial label complexity guarantees. Specifically, an example is provided to show that the proposed algorithm has superior label complexity over its passive version. The key technical trick is to select an appropriate "scale", t, such that the margin error term epsilon(t), and the sample compression generalization bound phi(t), are almost of the same order. After selecting the scale, a 1-NN classifier is output.

Qualitative Assessment

Technical quality: the proofs in this paper are quite sound. For practical implementation, I wonder if the labels queried for different scales t can be reused. Perhaps we can organize the data in some tree structure, e.g. cover trees (Beygelzimer, Kakade, Langford, 2004), and walk down the cover tree in some way. In addition, I wonder if the idea of "active comparison of pairs of classifiers" in (Balcan, Hanneke, Vaughan, 2010) can be used in the model selection (scale selection) procedure, since the term O(1/\hat{G}) term is a bit unsatisfying to me. The fact that the label complexity bound is exponential in the doubling dimension of the metric space is a bit concerning, although I think this type of dependency may be difficult to get rid of for nonparametric classifiers. Novelty: although the general idea of using sample compression bounds for condensed nearest neighbor classification has appeared in e.g. (Gottlieb, Kontorovich, Nisnevich, 2014), the idea here using random sampling in each Voronoi cell and noisy binary search the "correct" scale to minimize the error bound is novel. Potential Impact: There has been relative little literature on active learning with non-parametric classifiers, e.g. nearest neighbor classifier. I think this is a nice starting point and will spur future work. Clarity and Presentation: The main result, Theorem 3.1 is a bit difficult to appreciate on first sight, since it is very different from PAC learning with a fixed/parameterized hypothesis class. In addition, the paper may need some explanation on how close is the error of the classifier to the Bayes optimal, in additive sense. I find Theorem 5.4 and 5.5 difficult to follow, but I understand the high level idea as follows -- we would like to find two neighbor scales t_0 < t_0' such that \hat{\epsilon}(t_0) < \phi(t_0) \hat{\epsilon}(t_0') > \phi(t_0') If this is true then the scale t_0 is optimal up to constants in terms of error bound achieved. Is that right? *** UPDATE AFTER READING AUTHOR FEEDBACK AND DISCUSSION *** I think overall this is a nice paper that provides active learning algorithm working in general metric space and providing nontrivial label complexity guarantees. I vote for acceptance. In the author feedback, the author says that in realizable case, Algorithm 1 of BHV10 is not directly applicable; I can roughly see that since the number of scales to be considered is O(m^2). But under some standard regularity assumptions (density for each class does not change too fast), can one use a "doubling trick", i.e. set scales t_L = t_1 \leq t_2 \leq .. \leq t_n-1 \leq t_n = t_U such that for any i, nu(t_i+1) \leq 2 * nu(t_i) phi(t_i-1) \leq 2 * phi(t_i) (*) and we make sure that nu(t_U) \leq 1/m phi(t_L) \leq 1/m (*) Now consider running Algorithm 1 of BHV10 with a label budget L. for each scale t, we call GenerateNNSet(t, I, delta) with label budget L/n. (If during the execution of the procedure it runs out of label budget then it simply returns a trivial classifier.) Then active comparison is performed to output a final classifier. This can make sure that the algorithm adapts to the "correct" scale with a additional multiplicative factor of O(n log n/\delta) label complexity. Admittedly, my idea is a crude one, since finding scales that satisfies conditions (*) apriori is a bit difficult without knowing anything about the data distribution. (Reviewer 1 agrees that something like this might work, but some technical details need to be worked out.)

Confidence in this Review

2-Confident (read it all; understood it all reasonably well)